# The Matthew Effect in Recovery from Smartphone Addiction in a 6-Month Longitudinal Study of Children and Adolescents [note 1]

**DOI:** 10.3390/ijerph17134751

**Published:** 2020-07-01

**Authors:** Seung-Yup Lee, Hae Kook Lee, Jung-Seok Choi, Soo-young Bang, Min-Hyeon Park, Kyu-In Jung, Yong-Sil Kweon

**Affiliations:** 1Department of Psychiatry, Eunpyeong St. Mary’s Hospital, College of Medicine, The Catholic University of Korea, Seoul 03312, Korea; seungyup@catholic.ac.kr (S.-Y.L.); neominnie@hanmail.net (M.-H.P.); cki@catholic.ac.kr (K.-I.J.); 2Department of Psychiatry, Uijeongbu St. Mary’s Hospital, College of Medicine, The Catholic University of Korea, Seoul 06591, Korea; nplhk@catholic.ac.kr; 3Department of Psychiatry, SMG-SNU Boramae Medical Center, Seoul 07061, Korea; psychoresi@hanmail.net; 4Department of Psychiatry, College of Medicine, Eulji University, Seoul 01830, Korea; dresme@daum.net

**Keywords:** problematic phone use, internet, pain, dry eye, depression, anxiety, quality of life, recovery, prognosis, cohort

## Abstract

The clinical course of problematic smartphone use (PSU) remains largely unknown due to a lack of longitudinal studies. We recruited 193 subjects with smartphone addiction problems for the present study. After providing informed consent, the subjects completed surveys and underwent comprehensive interviews regarding smartphone usage. A total of 56 subjects among the 193 initially recruited subjects were followed up for six months. We compared baseline characteristics between persistent addicted users and recovered users at the end of the 6-month follow-up. Persistent problematic smartphone users displayed higher baseline smartphone addiction severity and were more prone to develop mental health problems at the follow-up. However, baseline depressive or anxiety status did not significantly influence the course of PSU. PSU behaved more like an addictive disorder rather than a secondary psychiatric disorder. Harm avoidance, impulsivity, higher Internet use, and less conversation time with mothers were identified as poor prognostic factors in PSU. Lower quality of life, low perceived happiness, and goal instability also contributed to persistent PSU, while recovery increased these scores as well as measures of self-esteem. These findings suggest that the Matthew effect is found in the recovery of PSU with better premorbid psychosocial adjustment leading to a more successful recovery. Greater clinical resources are required for interventions in vulnerable populations to modify the course of this increasingly prevalent problematic behavior worldwide.

## 1. Introduction

Smartphone use has become an integral part of everyday life. While many conveniences are associated with widespread smartphone use, public health concerns regarding problematic smartphone use (PSU) have also increased over the past decade. In particular, lack of knowledge about the long-term consequences of PSU has raised further concerns in developmental studies of children and adolescents.

Even though connectivity is generally believed to be increased by smartphones, people may also snub others by touching their smartphones rather than engaging in person. This novel type of social exclusion phenomenon is called “phubbing” and was reported to be linked to social anxiety [1]. Thus, we can understand that novel problems may arise following novel technologies.

South Korea reported a smartphone penetration rate of 95% in 2018, the highest in the world [2]. In 2019, a nationally representative survey of South Koreans revealed PSU prevalence of 20.0% across all age groups [3], which corresponds to the global prevalence of 23.3% reported in a recent systemic review [4]. While many reports outline general difficulties in controlling excessive smartphone use, the groups at the highest risk in the 2019 Korean national survey were children and adolescents, with PSU prevalence of 22.9% and 30.2%, respectively [3].

There is contradictory evidence in the literature regarding whether smartphone addiction truly exists [5,6,7]. PSU is generally not regarded as a serious health issue like substance abuse. In a British survey, the majority of parents report that they do not regulate their children’s smartphone time [8].

PSU has been associated with depression, anxiety, aggression, decreased sleep duration/quality, and other addictive behaviors [4,9,10,11,12,13,14]. It is also related to physical health risks such as dry eye symptoms, musculoskeletal pain, and accidents [15,16,17,18,19,20]. While such physical problems may arise as a direct consequence of uncontrolled excessive smartphone use, the relationship between psychosocial problems and PSU is yet to be established. For instance, PSU was associated with depression and lower subjective happiness; attention deficit hyperactivity disorder (ADHD) and impulsivity; anxiety, and lower quality of life in cross-sectional studies [10,21,22]. However, it is yet unclear whether such negative mental and behavioral variables act mainly as risk factors that contribute to PSU development or such were derived from PSU as adverse outcomes. To complicate matters, factors such as self-esteem, goal instability, or parent–child relationships may also exert influence over the course of PSU among the developing children and adolescents. To verify their temporal order and relative contributions, well designed longitudinal studies are required.

However, longitudinal studies are scarce, and all existing longitudinal studies recruited convenient samples rather than surveying the clinical population [23,24,25,26]. The pre-existing longitudinal studies are limited by drawing conclusions exclusively by conducting surveys on non-clinical convenient samples. Non-clinical samples are more likely to have lower PSU severity and fewer related mental or physical health problems compared to clinical samples [27]. Despite being categorized as smartphone addiction, the severity of the subjects recruited from the non-clinical community settings may have not been severe enough to capture the true clinical picture of the population with clinically significant impairment by PSU.

Together with the lack of any longitudinal studies that demonstrate the diagnostic stability of PSU, the lower severity of the pre-existing studies due to their reliance on non-clinical samples may have contributed to the prior argument that PSU lacks health consequences severe enough to make it a legitimate addictive disorder [5]. Therefore, there is a strong need to investigate the clinical course of PSU in clinical settings and its diagnostic stability in the long term.

The lack of evidence regarding the long-term consequences of PSU is especially alarming since PSU is highly prevalent among children and adolescents [3]. The mental and behavioral consequences of PSU may not only deteriorate current function but may linger through the loss of future opportunities and altered psychosocial development. Furthermore, previous studies indicate that excessive Internet use heightens the risk of substance use [28], and problematic gaming elevates the future risk of developing gambling problems. Addictive use of social media was reported to be related to reduced well-being [29]. These findings demand closer attention to whether PSU occurring during this critical developmental period leads to mental health problems later [30].

The information technologies were suggested to potentially reduce social inequalities by increasing the reachability of information and educational resources to a wider population. However, the “digital divide” created by the disparity in access to online information may further broaden the social health inequalities of our society at the same time [31]. With the growing smartphone penetration around the globe, however, the inequalities by the lack of accessibility will likely decrease. In contrast, other forms of inequalities may emerge due to excessive smartphone use and its long-term consequences.

For instance, children from higher socioeconomic status were more likely to use their digital devices safely while punishing parental attitude was associated with increased risk of PSU [32,33]. While education is generally considered as the “great equalizer”, PSU was related to drug consumption and poor academic achievement [34]. The undermined educational opportunity in children and adolescents due to PSU may not only impact their lives in schools but lead to a lifelong socioeconomic disparity through employment and income across the generation by the missed opportunities in the crucial period of development.

“For to everyone who has, more will be given and he will grow rich; but from the one who has not, even what he has will be taken away.” (Matthew 25:29) [35]. The Matthew effect, a term derived from the Gospel of Matthew by Merton [36], will also be useful to describe the cumulative disadvantage that may potentially occur when PSU is not properly mitigated in children with individual and environmental vulnerabilities.

Therefore, in this study, we explore the course of PSU and prognostic factors that may influence its course. To the best of our knowledge, this is the first longitudinal study of PSU conducted in a clinical sample. In addition to smartphone-related variables, we performed follow-up measurements of psychosocial variables to determine their contributions to PSU. The identification of risk and prognostic factors related to PSU will facilitate the development of preventive and interventional strategies against this worldwide public health risk among children and adolescents.

## 2. Materials and Methods

### 2.1. Study Procedure and Participants

This study was conducted as a part of the clinical Cohort for Understanding of Internet addiction Rescue factors in Early life (c-CURE) study. C-CURE is a multi-center study involving three university hospitals in the Seoul metropolitan area of South Korea that explores risk and protective factors regarding harmful digital media use in children and adolescents. The sample includes subjects who visited the outpatient departments of these hospitals for the treatment of excessive Internet, game, or smartphone use.

After obtaining written informed consent from both participants and their parents/guardians, a brief 15–20 min session of parental coaching was conducted with the additional provision of a 12-page pocket reference outlining parental guidance on digital media use at baseline. The content of the pocket guide includes a parental checklist for children’s excessive use, explanation of risk factors (neglect, ADHD, depression, impulse control, and social phobia), explanation of game genres and motivations of children, parenting coaching tips, setting rules, fostering alternative recreational activities, and how to act when childrens relapse (checklist for warning signs, converse more with attentive listening, help children to find other activities that may interest them, provide emotional support, and seek expert’s help).

Of the initial 188 participants, we included 85 subjects (64.7% males and 35.3% females) aged between 7 and 18 years old (mean age of 13.2 years old). The inclusion criterion was demonstrating scores equal to or greater than 31 on the smartphone addiction scale-short version (SAS-SV) on the baseline assessment. The subjects were evaluated at three and six months of follow-up.

This study was conducted following the Declaration of Helsinki and was reviewed and approved by the Institutional Review Boards of Uijeongbu St. Mary’s Hospital, the Catholic University of Korea (UC15ONMI0072), Seoul Metropolitan Government, Seoul National University Boramae Medical Center (16 April 2016), and Eulji University Seoul Eulji Hospital (EMCS2015-05-020-001).

### 2.2. Measures

#### 2.2.1. Smartphone Addiction Scale-Short Version (SAS-SV)

This ten-item self-measurement was used to evaluate PSU. It is a shortened version of the original Smartphone addiction scale with 33 items that assess six components of addiction model—excessive use, tolerance, withdrawal, disturbance in daily-life, over-valued online relationship, and positive anticipation [37]. While the first four components correspond to diagnostic criteria of substance use disorder by the 5th edition of Diagnostic and Statistical Manual of Mental Disorders, the last two are unique to smartphone addiction scale.

The SAS-SV is rated on a six-point Likert scale (1 = strongly disagree to 6 = strongly agree) with the total scores ranging from 10 to 60. It questions items such as “Missing planned work due to smartphone use.” A higher score indicates greater severity of PSU, and a cutoff score of 31 was used in this study as previously suggested [38]. Cronbach’s alpha was 0.63 in this study.

#### 2.2.2. Internet Addiction Test (IAT)

We used the IAT to evaluate the severity of problematic Internet use. It questions items such as “How often do you choose to spend more time online over going out with others?” Rated on a 5-point Likert scale (1 = rarely to 5 = always), this 20 item self-measurement indicates more severe problematic Internet use with increasing scores. Cronbach’s alpha was 0.88 in this study.

#### 2.2.3. Survey on the Pattern of Digital-Media Use

Comprehensive surveys were conducted to evaluate digital media use. Participants answered questions about smartphone access such as starting age, average daily time spent on smartphones or the Internet, type of media content, and patterns of use such as bedtime smartphone usage. In addition, participants responded to questions about weight and height for calculating body mass index (BMI) and presence of musculoskeletal pain in the neck, shoulders, and hands/wrists/fingers to assess potential physical consequences of PSU.

#### 2.2.4. Dry Eye Symptoms (DES) Checklist

Questions about dry eye symptoms described jointly by the Korean ministry of health and welfare, the Korean Academy of Medical Science, and the Korean Ophthalmological Society were posed separately to the participants (foreign object sensation in the eyes, heaviness of the eyelid, eye-stiffness, blurry eyes, ocular fatigue, ocular soreness, frequent eye redness, dryness, photosensitivity, ocular pain, and string-like ocular discharge) [39]. The total number of positive responses to the DES checklist was used to measure the severity of DES. Cronbach’s alpha was 0.82 in this study.

#### 2.2.5. Junior Temperament Character Inventory (JTCI)

We utilized a child and adolescent version of the Temperament and Character Inventory, which was developed based on the biopsychosocial personality model of Cloninger [40]. The JTCI has been previously validated in Korean [41]. JTCI inventory assesses the four temperament types (novelty seeking, harm avoidance, reward dependence, and persistence) with yes/no responses to the 82 items. As high levels of novelty seeking and harm avoidance are well-known risk factors for addictive disorders such as gaming disorder [42], individual scores were anticipated to shift right in our study subjects. To explore the contributions of these variables within our study sample, individual scores were standardized to T-scores for further analyses.

#### 2.2.6. Depression Assessment

While considering participant age, the 27-item children’s depression inventory (CDI) was used to screen for depressive symptoms two weeks prior to assessment in children aged 12 or less. Each item is rated on a 3-point Likert scale from zero to two with descriptive statements such as “I feel sad sometimes, most, or always”. Participants with total scores of 22 or greater were classified as having depressive status, as previously suggested in a Korean validation study [43]. Cronbach’s alpha was 0.86 in this study.

The Beck Depression Inventory-II (BDI-II) was used to screen for depression in adolescents between 13 and 18 years of age. It contains five subscales of negative mood, interpersonal problems, ineffectiveness, anhedonia, and negative self-esteem. Each item is rated on a 4-point Likert scale from zero to three, and answers refer to the two weeks prior to assessment. It uses descriptions such as “I am so sad and unhappy that I cannot stand it” for each scale. Scores of 10 or greater on this 21-item self-questionnaire were used to identify participants with depression according to Beck [44]. Higher scores indicate more severe depressive symptoms in both tests. Cronbach’s alpha was 0.93 in this study.

#### 2.2.7. Anxiety Assessment

The Korean version of the State Anxiety Inventory for Children (SAIC), a part of the State-Trait Anxiety Inventory for Children, was used to assess anxiety in children aged 12 or less. This 20-item inventory includes questions regarding how frequently respondents feel worried, bothered, or nervous on a 3-point Likert scale (1 = almost never to 3 = almost always). With total scores ranging from 20 to 60, higher scores indicate greater anxiety. For this study, we used a score of 41 or higher as a cut-off for defining an anxious state, as previously suggested in a Korean validation study [45]. Cronbach’s alpha was 0.81 in this study.

The anxiety levels of adolescents between 13 and 18 years of age were examined with the State-Trait Anxiety Inventory-form X (STAI-X). Scores for this 20-item inventory range from 20 to 80 and the previously-defined cut-off point of 52 was used to classify subjects with high anxiety [46]. Cronbach’s alpha was 0.90 in this study.

#### 2.2.8. Barratt Impulsiveness Scale-11 (BIS-11)

The Korean version of the BIS-11, which includes three subcomponents of cognitive, motor, and non-planned impulsivity, was utilized to measure impulsivity [47]. This 23-item self-measurement is rated on a 4-point Likert scale (1 = rarely or never to 4 = almost always or always) with higher scores indicating greater impulsivity [48]. It questions items such as “I say things without thinking.” Cronbach’s alpha was 0.73 in this study.

#### 2.2.9. Conners-Wells’ Adolescent Self-Report Scale-Short Form (CASS-S)

We used the Korean version of the CASS-S, which is a 27-item instrument that assesses ADHD [49]. It is answered on a 4-point Likert scale (0 = not true at all to 3 = very often, very frequently). CASS-S has three subcomponents (conduct problems, cognitive problems, and hyperactivity), with higher scores suggesting more severe ADHD symptoms. CASS-S contains questions such as “I have difficulty sitting still.” Cronbach’s alpha was 0.83 in this study.

#### 2.2.10. Goal Instability

This inventory was developed from concepts derived from the self-psychology of Heinz Kohut that measure goal-directedness, self-competencies, and career decidedness [50]. Participants responded to the Korean version of the goal instability tool, which includes 10 items rated on a 6-point Likert scale [51]. It questions items such as “I have confusion about who I am.” Endorsement of each item results in higher scores, which indicate greater goal instability of the respondents. Cronbach’s alpha was 0.88 in this study.

#### 2.2.11. Adolescents Happiness Index (AHI)

We used the 30-item AHI to examine levels of positive emotion and well-being. This inventory is rated on a 4-point Likert scale (1 = strongly disagree to 3 = strongly agree) and has four sub-domains including self-concept, family relationship, leisure, and peer relationship [52]. Higher scores indicate higher levels of happiness. AHI asks questions such as “I am satisfied with myself.” Cronbach’s alpha was 0.92 in this study.

#### 2.2.12. Rosenberg Self-Esteem Scale (RSES)

Self-esteem was measured using a 10-item scale with each item rated on a 5-point Likert scale (1 = strongly disagree to 5 = strongly agree) [53]. With total scores ranging from 10 to 50, the RSES was developed to measure feelings of self-acceptance, self-respect, and positive evaluation of self. A higher score suggests better self-esteem. RSES contains questions such as “I am able to do things as well as most other people.” Cronbach’s alpha was 0.84 in this study.

#### 2.2.13. Pediatric Quality of Life (pedsQL)

The pedsQL^TM^ Generic Core 4.0 was used to evaluate the quality of life. This is a 23-item tool with responses on a 5-point Likert scale (conversions 0 = 100, 1 = 75, 2 = 50, 3 = 25, and 4 = 0 scores). The pedsQL has four subcomponents including physical, emotional, social, and academic functioning. Better quality of life is suggested by higher scores [54]. It questions items such as “hard to run” Cronbach’s alpha was 0.93 in this study.

### 2.3. Statistical Analyses

Fifty-six participants remained in the study among the 85 subjects included at baseline, for a follow-up rate of 65.9%. We grouped participants into a persistent group and a recovered group according to SAS-SV scores at the endpoint of 6-month follow-up. Participants whose SAS-SV scores were equal to or greater than 31 were classified in the persistent group, while those with SAS-SV scores less than 31 were considered recovered at the endpoint.

To identify prognostic factors of PSU, baseline sociodemographic and clinical variables between the two groups were compared. In addition, PSU-related variables and psychological variables were compared at 3- and 6-month follow-ups.

Chi-square or Fisher’s exact tests were performed for categorical variables. Mann–Whitney U-tests were conducted for continuous variables due to the limit of the final sample size at 6 months. To measure the effect size of PSU to variables at the follow-up, Cohen’s d, and Cramer’s V were calculated for continuous and categorical variables, respectively. All analyses were performed using SPSS Statistics for Windows, Version 24.0 (IBM Corp., Armonk, NY, USA) with two-tailed statistical significance set at *p* = 0.05.

## 3. Results

The baseline comparison between the PSU recovered and persistent PSU group was performed for the sociodemographic, smartphone-related, psychological, and physical measures to identify the prognostic factors. In addition, we compared the significant variables identified by the analyses at the baseline at the 3- and 6-month follow-ups to explore the clinical course.

There were no significant demographic differences between the recovered group and the persistent group including age, sex, or socioeconomic status (Table 1). However, the persistent group had fewer conversations with their mothers on weekdays (*p* = 0.002). The persistent group also showed significantly higher PSU at baseline as indicated by SAS-SV scores (*p* = 0.043), the average daily amount of time for mobile/smartphone use on both weekdays (*p* = 0.026) and weekends (*p* = 0.037), and Internet use time on weekdays (*p* = 0.044) (Table 1).

The online content consumed by the groups differed. The persistent group consumed more adult material, but the difference between groups was not statistically significant. The recovered group was more likely to be involved in blogging (*p* = 0.035).

In the comparison of psychological variables, the two groups did not differ significantly regarding the history of psychiatric disorders, current depression, or current anxiety. However, when stratified by age, younger children (less than 13 years of age) in the persistent group showed significantly higher SAIC scores (*p* = 0.035). Furthermore, the persistent group tended to display higher CDI, but the difference between groups was not significant (*p* = 0.06). While the persistent PSU group had significantly higher levels of harm avoidance (*p* = 0.014), BIS-11 (*p* = 0.036), and goal instability (*p* = 0.008), they had significantly lower pedsQL (*p* < 0.001) and AHI (*p* = 0.006) in the whole group comparison (Table 2).

When examining the physical consequences of PSU, baseline variables such as BMI, DES, shoulder, and hand/wrist/finger pain did not differ between the two groups. However, the symptom presentation of neck pain was significantly higher in persistent problematic users (*p* = 0.009) (Table 2).

In follow-up comparisons of variables that showed significant differences at baseline, BIS-11 and weekday conversation time with the mother did not demonstrate significant differences at follow-up visits. However, the recovered group continued to demonstrate higher pedsQL at both 3- and 6-month follow-ups (*p* < 0.001 for both) and also showed greater AHI at 6-month follow-up (*p* = 0.003). In contrast, the persistent PSU group showed higher goal instability (*p* = 0.011) and neck pain (*p* = 0.043) at the endpoint (Table 3).

The two groups continued to display significant differences in SAS-SV and the daily average amount of time for mobile/smartphone use on both weekdays and weekends, with a steeper reduction in the recovered group. IAT scores, which did not differ between groups at baseline (*p* = 0.371), became significantly different at both 3- and 6-month follow-ups (*p* = 0.006, and *p* < 0.001, respectively) (Figure 1).

We also conducted analyses at follow-up points of variables that were not significantly different at the baseline. The duration of bedtime phone usage before sleep was significantly lower in the recovered group at both 3- and 6-month follow-ups (*p* = 0.036, and *p* = 0.019, respectively). Recovered problematic users were significantly less likely to suffer from DES at 6-month follow-up (*p* = 0.019). The persistent group, however, was more likely to exhibit depressive and anxious states at 3- and 6-month follow-ups, respectively (Figure 2). RSES, which was not significantly different either at baseline or 3-month follow-up, was significantly different between groups at 6-month follow-up (*p* < 0.001) (Figure 3).

## 4. Discussion

PSU is increasingly prevalent worldwide. Depression, anxiety, sleep disturbance, poor academic achievement, and physical hazards such as increased risk of accidents, musculoskeletal problems, and ophthalmic problems have been reported in numerous cross-sectional studies [11,12,14,15,16,17,18,19,20]. However, the long-term consequences of PSU remain largely unknown. The ever-increasing penetration of smartphones among children and adolescents is particularly alarming since the long-term consequences of PSU on developmental trajectories remain unknown. In addition to direct adverse outcomes on mental or physical health, severely affected children may suffer indirectly by loss of future opportunities through the impoverished acquisition of knowledge or skills during critical periods of development.

The objective of this study was to identify prognostic factors of PSU among children and adolescents. Currently, there are ongoing debates concerning excessive smartphone use, whether it should be considered “addiction” or merely “problematic use” and whether such behavior is “primary” or “secondary”. Therefore, we also attempted to explore the temporal relationships between PSU and psychiatric problems, in addition to exploring the clinical course of PSU.

While there were no significant differences between the recovered group and the persistent PSU group in terms of psychiatric diagnosis, depression, or anxiety scores at baseline (Table 2), the persistent problematic smartphone users displayed a significantly higher rate of depression at 3-month follow-up and a significantly higher rate of anxiety at 6-month follow-up (Figure 2). Therefore, our findings support the designation of PSU as a primary disorder rather than a secondary phenomenon following negative emotional experiences in the clinical population. On the contrary, greater depressive or anxiety symptoms displayed by the PSU persistence suggest that adverse psychiatric consequences may develop unless PSU is adequately managed.

Although no significant differences were found in the whole group comparison, the anxiety level was significantly higher in children with persistent PSU aged 12 or less. Furthermore, although the difference was not statistically significant, a similar tendency was observed for depressive status in this younger group. These findings suggest that smartphones may serve as a compensatory measure against negative emotional experiences in younger children. Thus, PSU may also occur as a secondary disorder in young children. However, to confirm such a hypothesis, these findings should be replicated in a larger sample. When PSU is reported in children, considering both age and the context of use may help clinicians to understand the relationship between PSU and mental health.

The psychosocial vulnerabilities associated with persistent PSU at baseline were less conversation time with mothers, goal instability, low quality of life, lower level of perceived happiness, harm avoidance, and impulsivity (Table 2).

Parents exert significant influence over the development of their children. Parental perception of the social media use and parent–child interaction insecure attachment have been suggested as a major risk factors for addictive disorders [55], and punishing parental attitude was previously reported to be related to PSU [33]. Less conversation time with mothers may indicate weaker mother–child relationships. The lack of maternal protective factor was found to significantly increase the risk of continued problematic use of smartphones in this study. Despite not reaching a statistically significant level (*p* = 0.065), the persistent group was more likely to be living without their mothers, and it is in line with the aforementioned finding. Our finding is in line with the previous literature. In a large study of Chinese adolescents, mother–child relationships exerted more significant influence on problematic internet use than father–child relationships, and separated or “left-behind adolescents” were associated with subcomponents of problematic internet use [56].

The persistent PSU group showed lower quality of life and happiness but greater goal instability at baseline as well as at follow-up (Table 3). Further studies are indicated to investigate whether recovery from PSU by active therapeutic intervention also improves the quality of life, perceived happiness, and goal instability.

In short, the likelihood of recovery from PSU was higher among happy children with less goal instability, who interacted more with their mothers and enjoyed a better quality of life. The recovered users also spent significantly less time using smartphones at follow-up. Rosenberg Self-Esteem Scale scores, which demonstrated no significant differences at baseline, significantly increased at the endpoint among recovered participants (Figure 3). This indicates that recovered participants also exhibited increased self-esteem at 6-month follow-up.

The Matthew effect, which describes a psychosocial phenomenon of deepening inequality in which the rich become richer while the poor become even poorer [36], is thereby seen in the recovery process of PSU. The chance of recovery from PSU was lower in children with psychosocial disadvantages. PSU is likely to further widen the gap between children with psychological vulnerabilities or unfavorable family backgrounds and those without. The persistence of PSU may exert a significant influence over the developmental trajectory of self-esteem. While the self-esteem of the recovered children is boosted by the recognition of their success in achieving the recovery and better psychosocial outcomes such as higher quality of life and subjective happiness, the children with persistent PSU are more likely to fall behind in other important areas such as socialization and academic performance. Therefore, greater clinical attention and more intensive resources should be offered to PSU subjects with higher risk factor burdens.

Harm-avoidant temperament and impulsivity are well-known risk factors of PSU and other addictive disorders [12,42,57,58]. These risk factors may act as biological vulnerabilities to PSU. A previous structural imaging study demonstrated a significantly lower gray matter volume of the orbitofrontal cortex, an important structure involved in impulse control, in subjects with PSU compared to normal smartphone users [59]. These findings indicate shared mechanisms underlying PSU and addictive disorders. Further neurobiological studies are required to confirm the potential mechanism. Even though the impulsivity scores were higher in the persistent group throughout the follow-up, statistical significance was not sustained in the follow-up unlike the baseline. Therefore, further studies with a large sample size are required to determine whether impulsivity as a trait has a prognostic value. Another finding suggesting the addictive nature of PSU is the demonstrated prognostic value of PSU severity. Persistent problematic smartphone users demonstrated significantly higher initial PSU severity and spent greater amounts of time using smartphones, findings that are in line with those of a previous longitudinal study [24]. The fact that PSU severity per se at baseline, but not mental health problems, exert significant influence on the clinical course highlights the possibility that addiction plays a key role in PSU.

When participants with PSU were followed for 6 months, the majority (62.5%) continued to demonstrate problematic patterns of smartphone use. This finding suggests that the clinical course of PSU is stable over time and that more active management than parental education is needed to modify its course. The demonstrated diagnostic stability of PSU suggests that it may not be a benign behavioral problem lasting only temporarily.

Comparisons between the recovered group and the persistent group in this study demonstrated various physical consequences of PSU. Those with persistent smartphone problems were more likely to display neck pain not only at baseline but also at 6-month follow-up, which is in line with previous findings in the literature [16,19]. Considering that shoulder and hand/wrist/finger pain did not differ between the two groups, the assessment of neck pain seems to be the most clinically useful musculoskeletal symptom to screen for PSU and to monitor as a sign of PSU recovery.

Although no significant difference was observed at the baseline, the persistent group also showed significantly higher DES at the endpoint (Figure 2). In cross-sectional studies, longer smartphone usage was reported to be associated with DES [15,18]. This study is the first longitudinal study to demonstrate the development of DES as a consequence of persistent PSU.

Another longitudinal study previously showed that smartphone owners sleep significantly less during follow-up than subjects who do not own smartphones [60]. Exposure to blue light from smartphones at night was associated with disruptions of circadian rhythms and decreased performance at attentional tasks [61]. We found that the recovered group spent significantly less time using smartphones at bedtime at 3- and 6-month follow-ups. Therefore, bedtime smartphone usage should be evaluated in subjects with PSU to better manage potential sleep problems or difficulty concentrating in the daytime.

The two groups in this study were compared in the context of online content consumption. The consumption of adult material tended to be higher in the persistent group, but the difference was not statistically significant. However, the recovered group was more likely to engage in blogging. Still, it is too early to conclude that blogging is protective against PSU persistence. Maintaining a blog often involves sharing information such as the bloggers’ point of view, and it usually requires more time to write a blog than to write on social networking services such as Instagram. Therefore, bloggers may display lower harm avoidance or impulsivity. Further studies are needed to verify whether recovery is facilitated by such individual traits or actually by the maintenance of blogs.

To summarize, the results of our study support the application of the addiction model to PSU. PSU severity but not psychiatric comorbidity increased the risk of persistent PSU. Moreover, children and adolescents were more likely to develop physical and mental health problems with persistent problematic smartphone use. PSU not only showed characteristics of a primary disorder but also demonstrated diagnostic stability. As PSU shares common risk factors with addictive disorders such as impulsivity and harm avoidance, it manifests characteristics of addiction. However, it was previously argued that PSU is not an addictive disorder [5].

Several factors may have contributed to the traditional view. First, the traditional view was drawn from studies that utilized convenient samples and cross-sectional designs. To the best of our knowledge, the present study is the first longitudinal study conducted in clinical settings. The severity of problematic smartphone use in the non-clinical population is likely to be lower than in the clinical population, with higher self-remission. Therefore, the benign nature of PSU in the non-clinical population may have played a role in the formulation of such a benign view. In addition to differences in severity, the wide use of smartphones in everyday life may also make researchers more reluctant to label PSU as addictive behavior, in a desire to avoid criticism due to pathologizing and medicalizing daily behaviors. Even though we are also against pathologizing the simple overuse of smartphones by referring to it as an addictive disorder, PSU may have a wide spectrum in terms of severity and longitudinal course. Heavy users at the end of the continuum may display clinical characteristics resembling those of addicts. For such users, broader management options should be available, and the allocation of greater therapeutic resources will be required to improve clinical outcomes.

The strength of our study lies in its comprehensive assessment of demographic, psychosocial, and digital media-related risk factors. However, this study also has some limitations. While the majority of participants were followed throughout the study period, a substantial number (34.1%) dropped out at the 6-month follow-up. We conducted secondary analyses comparing the completed group and dropouts. There were no significant differences in demographics between groups. However, the baseline SAS-SV scores of the dropouts were significantly lower compared to the participants that remained in the study. Subjects with lower PSU severity may have been less motivated to continue the study.

Although our sample size is relatively large compared to those of other longitudinal studies, we were unable to perform parametric analyses for continuous variables due to the sample size. Lack of such analyses generally leads to a loss in statistical power, and small potential differences that otherwise might have been detected in parametric analyses may not have been found in the non-parametric analyses. Another limitation is the lack of objective measurements to validate the responses of the participants. Although the participants were reassured regarding the confidentiality of assessments to parents or guardians unless immediate harm was anticipated, the reliance on self-reports may result in an under-estimation of problems or severity. Significant discordance was reported between self-reported and clinician-rated severity of problematic gaming in a previous study [62]. Social desirability is known to contribute to under-reporting of deviant or socially undesirable behaviors [63]. Therefore, the utilization of objective measures such as smartphone applications that analyze patterns of use is required to manage the potential for under-reported PSU in children and adolescents.

## 5. Conclusions

In this longitudinal study, diagnostic stability was observed for PSU, with the majority of participants displaying persistent PSU six months after the beginning of the study. Prognostic factors of PSU were individual traits such as harm avoidance and impulsivity; psychological vulnerabilities such as high SAS-SV, low quality of life, low perceived happiness, and goal instability; and social risk factors such as shorter weekday conversations with mothers. We suggest that the bio-psychosocial framework could be useful in the evaluation of PSU and in predicting its course. The exploration of causality between PSU and psychiatric problems showed that PSU increased the risk of developing psychiatric problems later, which supports the interpretation of PSU as a primary disorder. While the recovered group demonstrated fewer psychological risk factors at baseline than persistent problematic users, they also benefited from further gains in psychological well-being and were less afflicted by physical consequences. This Matthew effect in the recovery of PSU calls for additional attention to be paid to disadvantaged children or adolescents with PSU, to better facilitate their recovery and reduce the harmful long-term consequences of PSU as a secondary prevention strategy.

## Figures and Tables

**Figure 1 ijerph-17-04751-f001:**
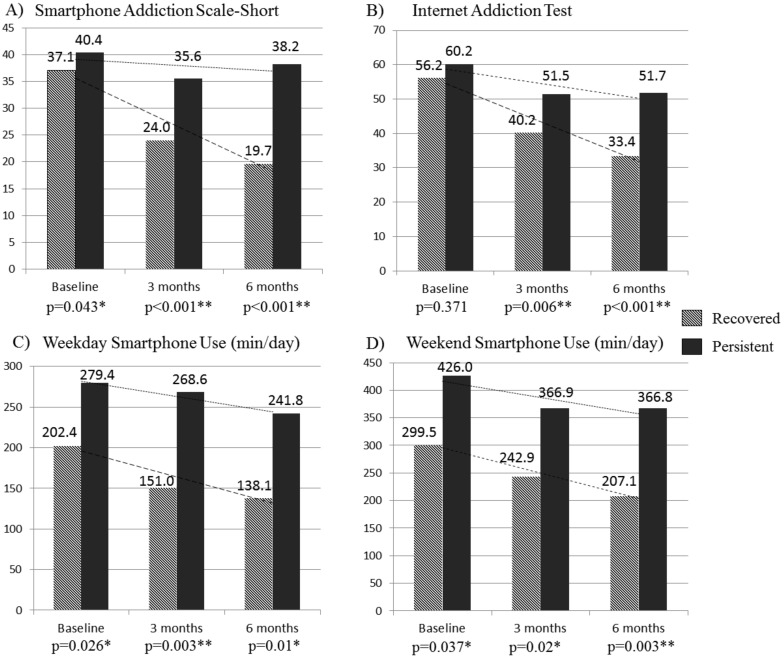
Longitudinal course of smartphone problems by recovery status. * *p* < 0.05, ** *p* < 0.01.

**Figure 2 ijerph-17-04751-f002:**
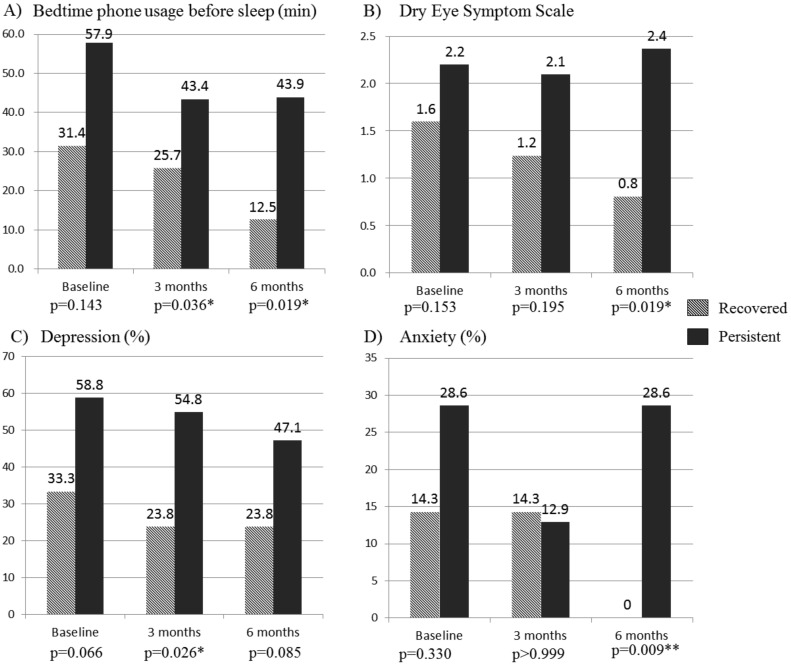
Physical and mental consequences of problematic smartphone use. * *p* < 0.05, ** *p* < 0.01.

**Figure 3 ijerph-17-04751-f003:**
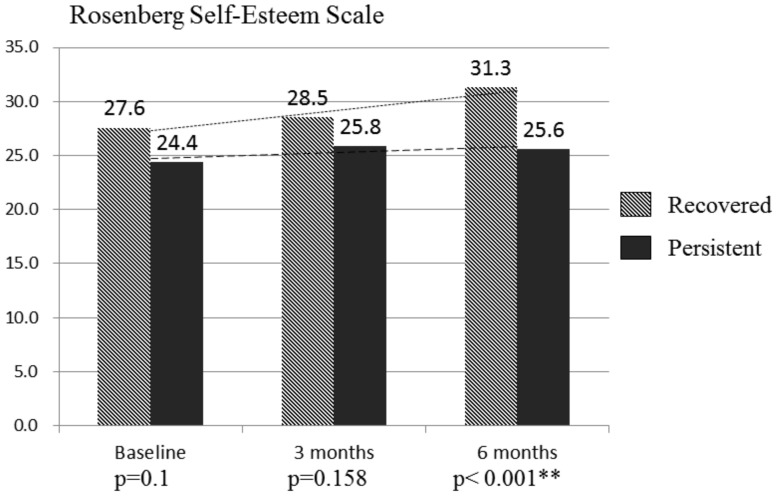
Follow-up comparison of self-esteem between the recovered and the persistent problematic smartphone users. ** *p* < 0.01.

**Table 1 ijerph-17-04751-t001:** Baseline comparison of sociodemographic and smartphone-related measures between recovered and persistent groups.

	Recovered Group (*n* = 21)	Persistent Group (*n* = 35)	*p*
Age (years)	12.2 ± 3.0	13.7 ± 2.5	0.083
Sex			0.822
Male	15 (71.4%)	24 (68.6%)	
Female	6 (28.6%)	11 (31.4%)	
Education status of mother			0.288
High school or lower	13 (61.9%)	22 (75.9%)	
College or higher	8 (38.1%)	7 (24.0%)	
Subjective economic status ^†^			0.537
Low	7 (36.8%)	11 (35.5%)	
Middle	6 (31.6%)	6 (19.4%)	
Upper-middle or above	6 (31.6%)	14 (45.2%)	
Living without father	7 (33.3%)	12 (35.3%)	0.882
Living without mother	0 (0%)	5 (14.7%)	0.065
Conversation time with mother (min)			
Weekday (average)	150.7 ± 166.6	71.7 ± 92.7	0.002 **
Weekend (average)	215.5 ± 327.1	142.8 ± 264.0	0.092
Smartphone use start (age)	8.5 ± 1.5	9.2 ± 2.3	0.369
Ownership of smartphone (+) ^†^	20 (95.2%)	34 (97.1%)	>0.999
Smartphone Addiction Scale-Short	37.1 ± 5.6	40.4 ± 6.0	0.043 *
Internet Addiction Test score	56.2 ± 14.9	60.22 ± 14.7	0.371
Mobile/smartphone use time (min)			
Weekday (average)	202.4 ± 111.8	279.43 ± 146.9	0.026 *
Weekend (average)	299.5 ± 137.5	426.00 ± 226.2	0.037 *
Internet use time (min)			
Weekday (average)	211.4 ± 84.0	282.9 ± 132.2	0.044 *
Weekend (average)	352.9 ± 158.7	444.0 ± 200.2	0.108
Internet content use			
News	9 (42.9%)	11 (31.4%)	0.388
Adult material	3 (14.3%)	12 (34.3%)	0.102
Online game	13 (61.9%)	26 (74.3%)	0.329
Blog	9 (42.9%)	6 (17.1%)	0.035
Social networking service	14 (66.7%)	20 (57.1%)	0.480
Bedtime phone use before sleep (min)	31.4 ± 38.2	57.86 ± 61.1	0.143

^†^ Fisher’s exact test; * *p* < 0.05, ** *p* < 0.01.

**Table 2 ijerph-17-04751-t002:** Baseline comparison of psychological and physical measures between recovered and persistent groups.

	Recovered Group (*n* = 21)	Persistent Group (*n* = 35)	*p*
*Psychological Assessment*			
Novelty seeking (T-score)	52.1 ± 8.3	52.0 ± 10.3	0.658
Harm avoidance (T-score)	46.8 ± 12.0	55.8 ± 10.2	0.014 *
History of Psychiatric diagnosis ^†^			
Depression (+)	2 (9.5%)	8 (22.9%)	0.290
Attention deficit hyperactivity disorder (+)	4 (19.0%)	7 (20.0%)	>0.999
Depression (+) status	7 (33.3%)	20 (58.8%)	0.066
CDI (n = 14) ^∮^	10.6 ± 4.8	18.0 ± 6.9	0.060
BDI (n = 41) ^∮^	17.3 ± 17.9	18.7 ± 12.8	0.306
Anxiety (+) status ^†^	3 (14.3%)	10 (28.6%)	0.330
SAIC (n = 14) ^∮^	27.0 ± 3.5	35.2 ± 2.9	0.002 **
STAI-X (n = 42) ^∮^	43.2 ± 14.9	47.4 ± 13.4	0.355
Barratt Impulsiveness Scale-II	56.1 ± 9.1	61.0 ± 6.9	0.036 *
Goal instability	29.9 ± 13.0	38.9 ± 10.8	0.008 **
CASS-Short form	25.9 ± 9.8	28.8 ± 10.6	0.436
Rosenberg Self-Esteem Scale	27.6 ± 6.4	24.4 ± 5.4	0.100
Quality of life	1911.9 ± 350.3	1472.1 ± 420.4	<0.001 **
Happiness scale	60.1 ± 16.1	49.1 ± 12.9	0.006 **

*Physical Assessment*			
Body mass index	22.6 ± 5.0	21.9 ± 5.9	0.264
Dry Eye Scale	1.6 ± 2.2	2.2 ± 2.1	0.153
Musculoskeletal pain (+)			
Neck	2 (9.5%)	15 (42.9%)	0.009 **
Shoulder ^†^	1 (4.8%)	9 (25.7%)	0.072
Hand/Wrist/Finger ^†^	2 (9.5%)	7 (20.0%)	0.459

^†^ Fisher’s exact test, ^∮^ Stratified by age as CDI and SAIC ≤ 12; BDI and STAI-X > 12; * *p* < 0.05, ** *p* < 0.01. CDI, Children’s Depression Inventory; BDI, Beck Depression Inventory; SAIC, State Anxiety Inventory for Children; STAI-X, State-Trait Anxiety Inventory form X; CASS, Conners-Wells’ Adolescent Self-Report Scale.

**Table 3 ijerph-17-04751-t003:** Follow-up findings of the recovered and persistent groups.

		Recovered Group (*n* = 21)	Persistent Group (*n* = 35)	*p*	Cohens’ d
Barratt Impulsiveness Scale-II	3 months	54.3 ± 11.1	57.0 ± 9.3	0.373	0.26
	6 months	53.0 ± 10.1	57.0 ± 8.5	0.1	0.43
Goal instability	3 months	28.2 ± 10.6	33.9 ± 11.5	0.073	0.52
	6 months	25.5 ± 12.1	34.7 ± 13.0	0.011 *	0.73
Quality of life	3 months	1994.1 ± 246.7	1591.7 ± 405.9	<0.001 **	1.2
	6 months	2078.6 ± 271.7	1612.1 ± 451.7	<0.001 **	1.25
Happiness scale	3 months	60.9 ± 16.8	52.0 ± 16.1	0.077	0.54
	6 months	60.7 ± 11.1	50.0 ± 14.2	0.003 **	0.84
					Cramer’s V
Neck pain	3 months	3 (14.3%)	8 (24.2%)	0.691 ^†^	0.17 (*p* = 0.46)
	6 months	2 (10.0%)	10 (35.7%)	0.043 *	0.29 (*p* = 0.04)

^†^ Fisher’s exact test; * *p* < 0.05, ** *p* < 0.01.

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
