# Peer review of "The Matthew Effect in Recovery from Smartphone Addiction in a 6-Month Longitudinal Study of Children and Adolescents [Author-notes fn1-ijerph-17-04751]"

_ijerph, 2020, doi:10.3390/ijerph17134751_

Round 1
Reviewer 1 Report
Thank you for the opportunity to review the manuscript entitled, “The Matthew effect in recovery from smartphone addiction in a 6-month longitudinal study of children and adolescents”.
I believe this study investigated a topic relevant to the readers of “IJERPH”. The use of Information Technology and Communications (ICT) is an essential aspect of modern societies. Access to such tools is increasingly easy and their use is not problem-free. Among ICTs, the mobile phone (smartphone) is the most popular. The intensive use of smartphones has been a cause of concern to researchers and institutions alike. While their use does not in itself pose a problem, the problematic relationship established with them does because their use over a large number of hours a day or in an uncontrolled manner can condition social relationships and the mental health. The children and adolescents population are the age groups considered to be at highest risk. The mobile phone is of much greater importance for them than for other age groups.
This paper is well written and follows well accepted standards of academic writing. Strengths include originality, interest and importance of the study. However, minor revisions may prove beneficial.
The introduction is very simply, not analyze in detail the relationship between impulsivity, happiness, quality of life, goal instability… and problematic smartphone use (PSU). The introduction lacks a clear objetives of the investigation. There is no relationship between the introduction and the variables that are measured
Participants: is need the percent of boy and girls, the age and the average age...
The manuscript lacks a clear study design. The design is not the procedure.
The instruments must appropriate to the research questions. The instruments must always display two important qualities: reliability and validity. Is need the alpha (α), Composite Reliability (CR) and Average Variance Extracted (AVE) of the Scales and Subscales.
Author Response
Thank you very much for the review
Here we attach our responses. Please see the attachment.

Reviewer 2 Report
The manuscript “The Matthew effect in recovery from smartphone addiction in a 6-month longitudinal study of children and adolescents” by Lee et al. reports an analysis on factors influencing the outcome in problematic smartphone use in children and adolescents. Among factors differing between recovered and persistent subjects, interaction with mother, impulsiveness, goal instability, quality of life and happiness were identified.
The study is interesting, well designed and clearly reported. Suggestions to improve the manuscript:
Lines 77-78: this sentence is not clear, it apparently states both that information technologies are supposed to increase and decrease inequalities, please reformulate.
Lines 96 and 286: the word “causal” seems too strong for association studies.
Line 110: It should be better explained how were the 85 subjects selected, whether they were all subjects that corresponded to the cut-off score or if there was any other selection criterion.
Line 228 (and discussion): my suggestion is to rephrase: if the difference is not significant, it means that as far as we know, there is no difference. This is especially true for online gaming, which is very far away from cut-off p value.
In Table 2, when the variables were assessed in the entire group and when in stratified subgroups should be better indicated in the table legend.
Impulsiveness scale score seems a personality trait, which would be unlikely to change in such a brief time-frame. Therefore, it is conceivable that the change at baseline is a false positive. Can the authors comment on this possibility?
Since state anxiety inventory for children scores were significantly different at baseline, it would be interesting if also the results at 3 and 6 months were reported in stratifications, not only as aggregated results.
The finding in Table 1 that living without mother shows a weak signal with p= 0.065 is in line with what is discussed at lines 307-311, perhaps it could be mentioned.
Line 327: in my opinion, data suggest that recovered children increase their self-esteem, more than that persistent children reduces theirs; it can be speculated that the good result in the therapy and possible commendation by the family can be at the basis of this increased self-esteem.
Minor:
Line 57: “are limited” space is missing.
Line 69: period repeated at the end of the sentence.
Line 80: “another form of inequality” or “other forms of inequalities”.
Author Response
Thank you very much for the review.
We deeply appreciate how thoroughly you read and left wonderful comments.
Here we attach our responses. Please see the attachment.

Reviewer 3 Report
The authors investigated how PSU affected several measures longitudinally (e.g., smartphone use, quality of life, self-esteem, anxiety, depression). They also relied on a specific intervention which resulted effective for roughly 38% of participants.
Several issues need to be "fixed", before publication to improve the manuscript.
1) Phubbing is a quite novel construct that I suggest you to at least mention. It is linked to social anxiety and is related to many forms of tech-addiction. A recent phubbing model could be found here:
Guazzini, A., Duradoni, M., Capelli, A., & Meringolo, P. (2019). An explorative model to assess individuals’ phubbing risk. Future Internet, 11(1), 21.
2) The authors stated, "...another form of inequalities may emergy due to excessive smartphone use and its long-term consequences". This is true but some examples would be recommended. Also, I would like to point out that for a non-clinical population the relationship between addiction measures (like smartphones, social media, and so on) and well-being exists but it is very small.
Ref: Duradoni, M., Innocenti, F., & Guazzini, A. (2020). Well-Being and Social Media: A Systematic Review of Bergen Addiction Scales. Future Internet, 12(2), 24.
This aspect is very important and should be highlighted. Indeed, it could be conceived as a further justification for selecting a "clinical" sample.
3) Another point I would suggest regards the inclusion criteria. The authors used the smartphone addiction scale-short version (SAS-SV). To increase readers understanding of this aspect it would be appropriate to specify: 1) the addiction model to which the instrument has been built; 2) which aspects typical of the DSM addiction definition are encompassed in this instrument and which are not.
4) A major flaw in this paper is that no information is given about what the participants are actually exposed to during the 6-months period. We have access only to this string in the procedure section "a brief 15-20 minute session of parental coaching was conducted with the additional provision of a 12-page pocket reference outlining parental guidance on digital media use at baseline". Could the authors provide additional information about this aspect or at least the key elements contained in the 12-pages?
5) Additionally for each measure, item examples and psychometric properties are needed.
6) I wish the authors could include the motives for some analyses, like the Mann-Whitney U test. Where the data non normally distributed? It is not a big deal, but it could be useful to have this information for future research.
7) At the beginning of the result section, it would be useful to have a sort of brief introduction to what the authors are going to analyze and why. Moreover, for each analysis, the authors should present the statistic (e.g., the value of Chi-square) and the effect size if computable and for statistically significant results only. For instance, you can rely on Phi coefficient and/or Cramer's V to compute the effect size of the Chi-square.
8) Finally, I would like the authors to hypothesize why the same intervention was useful for a group (the recovered) and not for the others (the persistent) in a cleaner way, since as they stated: "There were no significant differences between the recovered group and the persistent PSU group in terms of psychiatric diagnosis, depression, or anxiety scores at baseline". What are the factors that make the persistent PSU group less prone to change their behavior? This aspect is crucial to be highlighted. The time spent by mothers seems to be a good possible call and should be highlighted even more since the intervention, as far as I understand it, is centered and promoted by family members.
In this sense, I can recommend the following work on parents' perception of social media usage.
Procentese, F., Gatti, F., & Di Napoli, I. (2019). Families and Social Media Use: The Role of Parents’ Perceptions about Social Media Impact on Family Systems in the Relationship between Family Collective Efficacy and Open Communication. International Journal of Environmental Research and Public Health, 16(24), 5006.
Author Response
Thank you for the review.
We tried to reflect as best as we can in response to the reviewer's suggestion.
Here we attach our responses. Please see the attachment.

Round 2
Reviewer 3 Report
The authors answered effectively to the revision, and the paper is currently acceptable for publication.